# Contrast Invariant Feature Representations for Segmentation and Registration of Medical Images

**Yue Zhi, Russ Chua**                                      RUSSCHUA@MIT.EDU
**Adrian Vasile Dalca**                                     ADALCA@MIT.EDU

## Abstract

Imaging tasks like segmentation and registration are fundamental in a broad range of medical research studies. These tasks are increasingly solved by machine learning based methods. However, given the heterogeneity of medical imaging modalities, many existing methods are not able to generalize well to new modalities or even slight variations of existing modalities, and only perform well on the type of data they were trained on. Most practitioners have limited training data for a given task, limiting their ability to train generalized networks. To enable neural networks trained on one image type or modality to perform well on other imaging contrasts, we propose `CIFL`: contrast invariant feature learning. CIFL uses synthesized images of varying contrasts and artifacts, and an unsupervised loss function, to learn rich contrast-invariant image features. The resulting representation can be used as input to downstream tasks like segmentation or registration given some modality available at training, and subsequently enables performing that task on contrasts not available during training. In this paper, we perform experiments that demonstrate generalizability in brain segmentation and registration.

**Keywords:** Segmentation, Registration.

## 1. Introduction and Related Work

Imaging technologies including photographs, Magnetic Resonance Imaging (MRI) and Computational Tomography (CT)-Scans have provided effective means of medical diagnosis and treatment. As a result, there is significant variability in medical images given a variety of acquisition technologies, vendors, protocol choices, and patient populations even within a single institution. This poses a problem for neuroimaging tools that are usually trained on very specific MRI pulse sequences in available data sets which poorly generalize to unseen MRI modalities at inference. To address this, data augmentation techniques are often employed to produce Convolutional Neural Networks (CNNs) that generalize across MRI pulse sequences. Some methods augment data with MRI-based forward models that leverage physics-domain approximations to generate plausible, synthetic training examples similar to MRI pulse sequences (Jog et al., 2019). Dense Cycle Generative Adversarial Neural Networks (GANs) achieve adaptation between image modalities by synthesizing one modality from another (Lei et al., 2019). Recent strategies learn to use *anatomically-consistent* spatial deformation fields and intensity augmentations in segmentation tasks (Chaitanya et al., 2019; Zhao et al., 2019). Synthetic generation techniques of unseen MRI contrasts were also explored (Billot et al., 2020; Hoffmann et al., 2021b; Hoopes et al., 2022). However, all these methods train networks specific to a certain anatomy or deep-learning task (e.g. either segmentation or registration). In this paper, we build on synthetic generation methods but focus on producing general feature representations that are invariant to image contrast (modality) and are useful for a variety of analysis tasks.

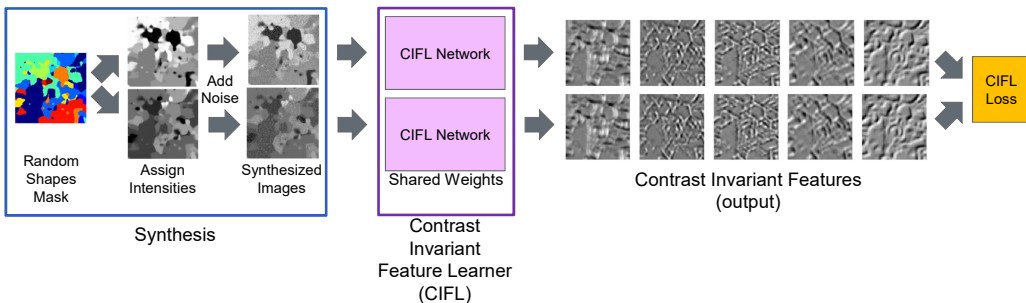

Figure 1: From a synthesized label mask of random shapes, we assign random intensities to each anatomy class to generate two different contrasts. We subsequently add random noise to produce a synthesized image. We apply a Contrast Invariant Feature Learner to each of the two images (with shared weights), giving image-sized feature representations. We use an unsupervised loss which encourages the features to be rich (diverse across channels of the representations) yet similar for the two contrasts.

## 2. Method

We define function $f_\theta : \mathbb{R}^{l \times w \times h} \to \mathcal{S}^{l \times w \times h \times C}$ with parameters $\theta$ that encodes a feature representation $r$ for input image $x$. The feature representation $r = f_\theta(x)$ is a $C$-channel image of the same spatial dimensions as $x$. For two images $x_{m_1}$ and $x_{m_2}$ of the same anatomy but different modalities $m_1$ and $m_2$, we encourage two properties for $r$:

- Similar representation of two modalities from the same anatomy: $f_\theta(x_{m_1}) \approx f_\theta(x_{m_2})$

- Rich representation to be usable in downstream applications: $f_\theta^c(x_{m_1})$ and $f_\theta^{c'}(x_{m_1})$ should be different, where the superscript $c$ represents the $c$-th channel of the feature representation.

To achieve these properties, we build on contrastive learning to optimize the loss

$$\mathcal{L}_{\text{contrastive}}(\theta, \tau_\alpha, \tau_\beta, \tau_\gamma; \mathcal{X}) \triangleq$$
$$\mathbb{E}_{(x_{m_1}, x_{m_2})} \left[ -\log \frac{\sum_c e^{f_\theta^c(x_{m_1}) \cdot f_\theta^c(x_{m_2})/\tau_\alpha}}{\sum_c e^{f_\theta^c(x_{m_1}) \cdot f_\theta^c(x_{m_2})/\tau_\beta} + \sum_c \sum_{c' \neq c} e^{f_\theta^c(x_{m_1}) \cdot f_\theta^{c'}(x_{m_1})/\tau_\gamma}} \right],$$

where $\tau_\alpha$, $\tau_\beta$ and $\tau_\gamma$ are individual temperature terms that scale their effects and $\mathcal{X}$ is a dataset of multi-modality images. We employ synthetic images of different shapes and contrasts to train such a network following SynthMorph (Hoffmann et al., 2021a). This is illustrated pictorially in Figure 1 which provides a broad overview of our CIFL training process. We employ a CNN to approximate $f_\theta$ with eight convolutional layers each with kernel size 3 and normalize the output final CIFL features using a $\ell_2$ normalization layer to distribute features onto a unit hypersphere, producing uniform intensities and closer positive

Table 1: Performance (Dice Score) on downstream tasks, on modalities unseen during training.

| | | | Method Dice Score | | | |
| | | | CIFL | | | |
| Task | Dimension | Dataset | Baseline | $\tau_\gamma = 1$ | $\tau_\gamma = 0.1$ | $\tau_\gamma = 0.01$ |
|---|---|---|---|---|---|---|
| Segmentation | 2D | Inverted OASIS T1 | $0.18 \pm 0.01$ | $\mathbf{0.84} \pm 0.04$ | $0.83 \pm 0.04$ | $0.79 \pm 0.04$ |
| Registration | 2D | OASIS T1-Inverted T1 | $0.45 \pm 0.02$ | $\mathbf{0.69} \pm 0.05$ | $0.69 \pm 0.05$ | $0.66 \pm 0.06$ |
| | | IXI T1-T2 | $0.36 \pm 0.03$ | $0.59 \pm 0.13$ | $0.59 \pm 0.12$ | $\mathbf{0.61} \pm 0.13$ |

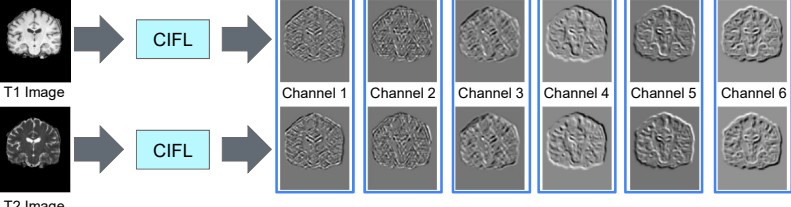

Figure 2: Example of CIFL features yielded by a trained CIFL network for two different MRI pulse sequences. The feature representations are similar for any two modalities in the same channel shown in each blue rectangle. The representations for each image are also different across channels (between blue rectangles), which provide downstream deep learning models with rich information.

alignments (Wang and Isola, 2020). To train standard downstream networks for segmentation and registration tasks (Balakrishnan et al., 2019; Ronneberger et al., 2015), we use the feature representations yielded from the trained CIFL network to generate downstream input features during inference on unseen modalities for the same task.

## 3. Experiments and Results

We perform preliminary experiments where we first train a CIFL network on image data of synthetic shapes. Then we train a downstream network using CIFL features from a T1 MRI brain image, and test the performance of those networks using CIFL features from unseen modalities. For downstream tasks, we employ the OASIS dataset (Hoopes et al., 2021; Marcus et al., 2007) of T1 images, processed to be normalized, affinely aligned and include 5-label segmentation maps consisting of the background, white matter, grey matter, cortical spinal fluid, and thalamus. In this preliminary work, we extract the mid-coronal slice and work in 2D. We partitioned 232 images for training, and 58 for validation. We then test each model on 100 samples of unseen images from OASIS, and Information eXtraction from Images (IXI) dataset[1]. Our preliminary experimental results in Table 1 show promise that the CIFL features can enable generalizability to unseen modalities shown in Figure 2.

---

1. IXI Dataset: https://brain-development.org/ixi-dataset/

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
