# OpenReview forum: "Contrast Invariant Feature Representations for Segmentation and Registration of Medical Images"
_MIDL.io/2023/Short_Paper_Track — MIDL 2023 Short paper track Poster_

### Official Review · Reviewer_mvAZ · 2023-04-23
**Interesting pretraining strategy but some aspects are not clear**

**Rating:** 6
**Confidence:** 4

**Review:**

The paper proposes a pre-training strategy based on contrastive learning that can be used for image-to-image translation tasks. The proposed strategy aims at learning image features that are agnostic to anatomy and modality by encouraging similar features of different modalities from the same anatomy and a diverse feature representation across the channels. However, it is unclear from the text how the feature diversity property is promoted using contrastive loss. This pre-training strategy is evaluated on image segmentation and registration and shows promising results.

---

### Official Review · Reviewer_KXEt · 2023-04-24

**Rating:** 7
**Confidence:** 5

**Review:**

This paper proposed CIFL (contrast invariant feature learning) noises to address the issue of data heterogeneity in image segmentation and registration problems. It uses explicitly contrastive learning on synthesized patterns in terms of intensities.

The idea and method to produce generalized contrast-invariant features are clearly described. It would be ideal to apply the proposed idea to 3D data rather than 2D. However, considering this is a short paper, it should be sufficient to demonstrate the idea.